# CHANNEL-DIRECTED GRADIENTS FOR OPTIMIZATION OF CONVOLUTIONAL NEURAL NETWORKS

## ABSTRACT

We introduce optimization methods for convolutional neural networks that can be used to improve existing gradient-based optimization in terms of generalization error. The method requires only simple processing of existing stochastic gradients, can be used in conjunction with any optimizer, and has only a linear overhead (in the number of parameters) compared to computation of the stochastic gradient. The method works by computing the gradient of the loss function with respect to output-channel directed re-weighted $H^0$ or Sobolev metrics, which has the effect of smoothing components of the gradient across a certain direction of the parameter tensor. We show that defining the gradients along the output channel direction leads to a performance boost, while other directions can be detrimental. We present the continuum theory of such gradients, its discretization, and application to deep networks. Experiments on benchmark datasets, several networks, and baseline optimizers show that optimizers can be improved in generalization error by simply computing the stochastic gradient with respect to output-channel directed metrics.

## 1 INTRODUCTION

Stochastic gradient descent (SGD) is currently the dominant algorithm for optimizing large-scale convolutional neural networks (CNNs) (LeCun et al. (1998); Simonyan & Zisserman (2014); He et al. (2016b)). Although there has been large activity in optimization methods seeking to improve performance, SGD still dominates in terms of its generalization ability. Despite SGD's dominance, there is still often a gap between training and real-world test accuracy performance, which motivates research in improved optimization methods.

In this paper, we derive new optimization methods that are simple modifications of SGD. The methods implicitly induce correlation in the output direction of parameter tensors in CNNs. This is based on the empirical observation that parameter tensors in trained networks typically exhibit correlation over output channel dimension (see Figure 1). We thus explore encoding correlation by constructing smooth gradients in the output direction, which we show improves generalization accuracy. This is done by introducing new Riemmanian metrics on the parameter tensors, which changes the underlying geometry of the space of tensors, and reformulating the gradient with respect to those metrics.

Our contributions are as follows. First, we formulate *output channel-directed* Riemannian metrics (a re-weighted version of the standard $\mathbb{L}^2$ metric and another that is a Sobolev metric) over the space of parameter tensors. This encodes channel-directed correlation in the gradient optimization without changing the loss. Second, we compute Riemannian gradients with respect to the metrics showing linear complexity (in the number of parameters) over standard gradient computation, and thus derive new optimization methods for CNN training. Finally, we apply the methodology to training CNNs and show the empirical advantage in generalization accuracy, especially with small batch sizes, over standard optimizers (SGD, Adam) on numerous applications (image classification, semantic segmentation, generative adversarial networks) with simple modification of existing optimizers.

### 1.1 RELATED WORK

We discuss related work in deep network optimization; for a detailed survey, see Bottou et al. (2018). SGD, e.g., Bottou (2012), samples a batch of data to tractably estimate the gradient of the loss function. As the stochastic gradient is a noisy version of the gradient, learning rates must follow

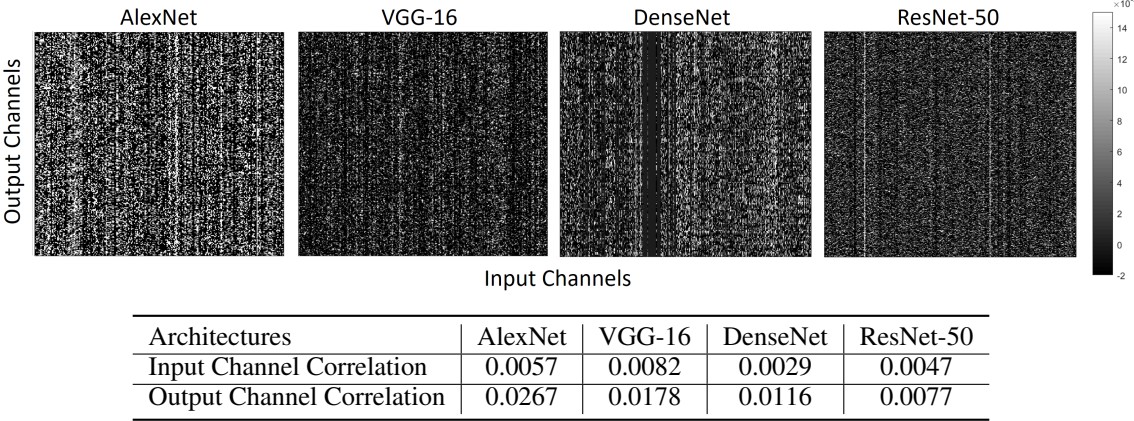

| Architectures | AlexNet | VGG-16 | DenseNet | ResNet-50 |
|---|---|---|---|---|
| Input Channel Correlation | 0.0057 | 0.0082 | 0.0029 | 0.0047 |
| Output Channel Correlation | 0.0267 | 0.0178 | 0.0116 | 0.0077 |

Figure 1: Visualization of parameter tensor of convolutional layers trained on ImageNet. Frequently within in layers (especially deeper layers), there is correlation of the weights along the output channel direction. The table shows output correlation (invariant to re-scaling) relative to input direction. Our method induces parameter correlations in the output direction.

a decay schedule in order to converge. Many methods have been formulated to choose learning rate over epochs and components of the gradient, including adaptive learning rates (e.g., Duchi et al. (2011); Zeiler (2012); Kingma & Ba (2014); Bengio (2015); Loshchilov & Hutter (2017); Luo et al. (2019)). For instance, Adam Kingma & Ba (2014) adaptively adjusts the learning rate so that parameters that have changed infrequently based on historical gradients are updated more quickly than parameters that have changed frequently. Another way to interpret such methods is that they change the underlying metric on the space on which the loss function is defined to an iso-tropically scaled version of the $\mathbb{L}^2$ metric given by a simple diagonal matrix; we change the metrics an-isotropically. We show that our method can be used in conjunction with such methods by simply using the stochastic gradient computed with our metrics to boost performance.

As the stochastic gradient is computed based on sampling, different runs of the algorithm can result in different local optima. To reduce the variance, several methods have been been formulated, e.g., Defazio et al. (2014); Johnson & Zhang (2013). We are not motivated by variance reduction, rather, inducing correlation in the parameter tensor to improve generalization. However, as our method smooths the gradient, our experiments show reduced variance with our metrics compared to SGD.

Another method motivated by variance reduction is Osher et al. (2018) (see applications Wang et al. (2019); Liang et al. (2020); Wang et al. (2020)), where the stochastic gradient is pre-multiplied with an inverse Laplacian smoothing matrix. For CNNs, the gradient with respect to parameters is rasterized in row or column order of network filters before smoothing. Our work is inspired by Osher et al. (2018), though we are motivated by correlation in the parameter tensor. Osher et al. (2018) can be interpreted as using the gradient of the loss with respect to a Sobolev metric. One insight over Osher et al. (2018) is that keeping the structure of the parameter tensor and defining the Sobolev metric with respect to the output-channel direction boosts accuracy, while other directions do not. Secondly, we introduce a re-weighted $H^0$ metric that preferentially treats the output-channel direction, and can be implemented with a line of Pytorch code, has linear (in parameter size) complexity, and performs comparably (in many cases) to our channel-directed Sobolev metric, boosting accuracy of SGD. Third, our Sobolev gradient, a variant of the ordinary one, has linear complexity rather than quasi-linear (not requiring FFT as Osher et al. (2018)). Sobolev gradients have been used in computer vision Sundaramoorthi et al. (2007); Charpiat et al. (2007) for their coarse-to-fine evolution Sundaramoorthi et al. (2008); we adapt that formulation to CNNs.

We formulate Sobolev gradients by considering the space of parameter tensors as a Riemannian manifold, and choosing the Sobolev metric on the tangent space. By choosing a metric, gradients intrinsic to the manifold can be computed and gradient flows are guaranteed to decrease loss. Other Riemannian metrics have been used for optimization in neural networks, e.g., Amari (1998); Marceau-Caron & Ollivier (2016); Hoffman et al. (2013); Gunasekar et al. (2020) and tangentially relate to our work. These works are based on Amari's Amari (1998) information geometry on probability measures, and the metric considered is the Fisher information metric. The motivation for these methods is re-parametrization invariance of optimization, whereas our motivation is imposing correlation in the

parameter space. Other works Gunasekar et al. (2020) use the Hessian metric (in the convex case), but these metrics are data-dependent and the gradient is challenging to compute, requiring (a large) inverse matrix computation.

## 2    CHANNEL-DIRECTED GRADIENTS

We now present the theory to define channel-directed gradients. To do this, we formulate new metrics on the space of tensors, and then derive analytic formulas for channel-directed gradients in terms of the standard $\mathbb{L}^2$ gradient. As we show, our channel-directed gradients effectively smooth the components of the $\mathbb{L}^2$ gradient across the output direction of the parameter tensors of the CNN, which induces correlation in that direction in the gradient and thus also the parameter tensor. Another interpretation is we are changing the geometry of the loss landscape (without changing the loss) to a more smooth one by changing the metric of the space on which the loss is defined.

### 2.1    BACKGROUND ON RIEMANNIAN GRADIENTS

We present the definition of gradient on a Riemannian manifold, and show the dependence of the gradient on the chosen metric on the manifold (see Carmo (1992); Abraham et al. (2012) for more details). A manifold $\mathcal{X}$ is a space that is locally linear around each point $X \in \mathcal{X}$; this linear space is the *tangent space*, denoted $T_X\mathcal{X}$. A *Riemannian manifold* has a smoothly varying positive definite bilinear form $\langle \cdot, \cdot \rangle$ (called the *metric*) on the tangent space. This metric allows one to define the notion of lengths of curves on the space, in addition to other operations, including gradients.

**Definition 1 (Gradient of a Function)** *Let $\mathcal{X}$ be a Riemannian manifold, and $f : \mathcal{X} \to \mathbb{R}$ be a function. The directional derivative of $f$ at $X \in \mathcal{X}$ along a direction $k \in T_X\mathcal{X}$ is defined as $\mathrm{d}f(X) \cdot k = \frac{\mathrm{d}}{\mathrm{d}\varepsilon} f(X + \varepsilon k)|_{\varepsilon=0}$. The* **gradient** *of $f$ at $X \in \mathcal{X}$ is the vector, $\nabla f(X) \in T_X\mathcal{X}$, that satisfies the relation*

$$\mathrm{d}f(X) \cdot k = \langle \nabla f(X), k \rangle, \text{ for all } k \in T_X\mathcal{X}. \tag{1}$$

Note that "the" gradient will depend on the choice of the metric on the manifold. We note that any such gradient will decrease the the function $f$ by moving infinitesimally in the tangent space in the direction of negative the gradient as $\mathrm{d}f(X) \cdot k = -\|\nabla f(X)\|^2 < 0$ when $k = -\nabla f(X)$, where $\| \cdot \|$ is the norm induced from the metric. The gradient flow, defined by the differential equation $\dot{X}_t = -\nabla f(X_t)$, will converge to a local minimum. In our application of this theory to CNN optimization, $f$ will be the loss function, and $\mathcal{X}$ will be the space of parameter tensors.

A consequence of this definition is that the gradient is the direction (up to a scale factor) in the tangent space that optimizes the following problem:

$$\underset{k \in T_X\mathcal{X} \backslash \{0\}}{\arg\max} \frac{|\,\mathrm{d}f(X) \cdot k\,|}{\|k\|}. \tag{2}$$

Thus, the gradient can be regarded as the most efficient direction as it maximizes the ratio of the change in energy by perturbing in a direction $k$ over the cost (defined by the metric) of $k$. Thus, by constructing the metric to have small costs for perturbations (directions) that we prefer for gradients, the gradient flow will move in these preferential directions while minimizng the function, and thus land in favorable local minima.

### 2.2    CHANNEL-DIRECTED METRICS

In existing deep network gradient-based optimization schemes, the underlying metric on the loss function is assumed to be the standard Euclidean $\mathbb{L}^2$ metric. We will consider a re-weighted version of the $\mathbb{L}^2$ metric and a Sobolev metric that favor correlation in the output channel direction of the gradient and thus the parameter tensors. To formulate the methodology, we start from a continuum formulation, where we treat weight tensors in the continuum, formulate the metrics in the continuum and then in the next sub-section derive the gradients with respect to these metrics. Finally, we discretize gradient flows in the implementation to derive iterative schemes.

Let $X : \mathcal{O} \times \mathcal{I} \times \mathcal{H} \times \mathcal{W} \to \mathbb{R}$ denote a parameter tensor of a layer of a convolutional neural network. Here $\mathcal{O} = [0, O]$ denotes indices to the output channel dimension of the tensor, $\mathcal{I} = [0, I]$

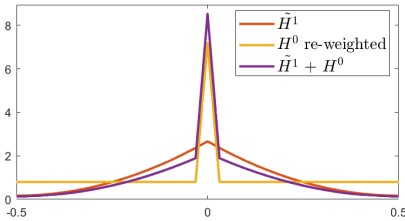

Figure 2: **Visualization of kernels applied to the $H^0$ gradient under different metrics for $\lambda = 1$.** This illustrates the smoothing effect of the metrics. In computation, linear cost formulas are applied to compute the gradients not using the convolution interpretation.

denote the indices to the input channel, and $\mathcal{H} = [0, H], \mathcal{W} = [0, W]$ denote indices to the height and width dimension of the spatial filters of the tensor. The metric is defined on the tangent space to the space of such $X$. An element of the tangent space will have the same form of the tensor, i.e., $k : \mathcal{O} \times \mathcal{I} \times \mathcal{H} \times \mathcal{W} \to \mathbb{R}$. The $\mathbb{L}^2$ (called $H^0$ from now on) metric is defined as

$$\langle k_1, k_2 \rangle_{H^0} = \int_{\mathcal{O}, \mathcal{I}, \mathcal{H}, \mathcal{W}} k_1(o, i, h, w) \cdot k_2(o, i, h, w) \, \mathrm{d}o \, \mathrm{d}i \, \mathrm{d}h \, \mathrm{d}w, \tag{3}$$

where $k_1, k_2$ are in the tangent space of tensors. We now define a re-weighted version of $H^0$ that favors tangent vectors that have global smoothness in the direction of the $\mathcal{O}$ dimension:

$$\langle k_1, k_2 \rangle_{H^0_\lambda} = \int_{\mathcal{I}, \mathcal{H}, \mathcal{W}} \bar{k}_1(i, h, w) \cdot \bar{k}_2(i, h, w) \, \mathrm{d}i \, \mathrm{d}h \, \mathrm{d}w + \frac{\lambda}{O} \left\langle k_1 - \bar{k}_1, k_2 - \bar{k}_2 \right\rangle_{H^0}, \tag{4}$$

where $\lambda > 0$ is a hyper-parameter, and $\bar{k}$ is the average value in the output channel direction, i.e.,

$$\bar{k}(i, h, w) = \frac{1}{O} \int_{\mathcal{O}} k(o, i, h, w) \, \mathrm{d}o. \tag{5}$$

The metric in (4) splits the tangent vector into global translations in the output channel direction and its orthogonal complement, i.e., the deformation. The weight $\lambda$ is used to control the weighting between the translation and deformation components, i.e., larger values of $\lambda$ means that deformations more heavily influence the norm of the perturbation. As shown in the next sub-section that means gradients with respect to this metric have higher weighted channel-directed translations than deformations.

Next, we introduce a channel-directed version of a Sobolev metric, defined as follows:

$$\langle k_1, k_2 \rangle_{\tilde{H}^1} = \int_{\mathcal{I}, \mathcal{H}, \mathcal{W}} \bar{k}_1(i, h, w) \cdot \bar{k}_2(i, h, w) \, \mathrm{d}i \, \mathrm{d}h \, \mathrm{d}w + \lambda O \left\langle \frac{\partial k_1}{\partial o}, \frac{\partial k_2}{\partial o} \right\rangle_{H^0}, \tag{6}$$

where $\frac{\partial}{\partial o}$ indicates the partial derivative with respect to the the output channel direction. The partial derivative in the $o$-direction implies that tensor perturbations that are smooth along the $o$-direction are close with respect to these metrics, which will imply that the corresponding gradients will exhibit smoothness in this direction, i.e., convolution filters that are nearby in the output direction will exhibit correlation. The metric is a weighted combination of the $H^0$ metric of the derivative in the output direction, and the $H^0$ metric of the output-directed translation. Note that the traditional Sobolev metric uses the $H^0$ metric of the perturbation rather than the translation. Our choice is motivated by computational efficiency of the corresponding gradient, to be discussed below. The scale factors of $O$ in the expressions above are so that the metric is scale invariant with respect to different sizes of output channels. The part of the metric with the partial derivative component implies that tensors that differ in the output channel direction by a non-smooth perturbation are far away in distance. Tensors that differ by just a channel-directed translation are close.

### 2.3 COMPUTING CHANNEL-DIRECTED GRADIENTS

We now compute gradients with respect to the metrics defined in the previous sub-section in terms of the $H^0$ gradient so that existing SGD code can be re-use with minimal changes. To compute the relation between the channel-directed gradients and the usual $H^0$ gradient, we note (1) that the directional derivative can be written as an inner product with the gradient with respect to any metric:

$$\mathrm{d}L(X) \cdot k = \left\langle \nabla_{H^0_\lambda} L(X), k \right\rangle_{H^0_\lambda} = \langle \nabla_{\tilde{H}^1} L(X), k \rangle_{\tilde{H}^1} = \langle \nabla_{H^0} L(X), k \rangle_{H^0}. \tag{7}$$

With this relation, we may compute the channel-directed gradients in terms of the $H^0$ gradient (details are in Appendix D). Letting $f = \nabla_{H^0} L(X)$, we have

$$\nabla_{H^0_\lambda} L(X) = \bar{f} + \frac{1}{\lambda}(f - \bar{f}) \quad \text{and} \quad f = \overline{\nabla_{\tilde{H}^1} L(X)} - \lambda O^2 \frac{\partial^2}{\partial o^2} \nabla_{\tilde{H}^1} L(X), \tag{8}$$

where the last expression is a second order ordinary differential equations (ODE), whose solution we discuss next. Notice that the re-weighted $H^0$ gradient (8) re-weights the channel-directed translation component and the deformation component of the $H^0$ gradient differently, i.e., as $\lambda$ gets larger, the channel-directed translation becomes more prominent.

Our Sobolev gradient effectively computes local averages, as we show, in the output channel direction, and by doing so effectively imposes an ordering of the kernels in CNNs so that nearby kernels (according to the distance in the output direction) are similar. As ordering of kernels in CNNs is arbitrary, in the sense that permutations of kernels in the output direction along with the input channels result in the same output, we are free to impose one ordering, which Sobolev effectively does during the optimization so that filters that are close in the $o$-dimension are similar.

In obtaining the expression for the Sobolev gradient below, we have assumed periodic boundary conditions in the $\mathcal{O}$ dimension, which further imposes the ordering of filters such that starting and ending filters in the $o$-dimension are similar. The periodic assumption gives that the Sobolev gradient can be computed with a circular convolution with the $H^0$ gradient, which is simpler to compute in practice. In fact, the Sobolev gradient is given as $\nabla_{\tilde{H}^1} L(X)(o, i, h, w) =$

$$\frac{1}{O} \int_{\mathcal{O}} K((o - \tilde{o})/O) f(\tilde{o}, i, h, w) \, \mathrm{d}\tilde{o}, \text{ where } K(o) = 1 + \frac{o^2 - o + 1/6}{2\lambda}, \text{ for } o \in [0, 1]. \quad (9)$$

Note that the re-weighted $H^0$ solution also has an interpretation of convolution with respect to a smoothing kernel. Figure 2 shows plots of the kernels for the parameter $\lambda$ chosen in experiments. For each $o$, the Sobolev or re-weighted $H^0$ is a local average whose weights die far away from $o$. Thus, the effect of the metrics is to induce smoothness of the gradient along the output channel direction.

The Sobolev gradient need not use the convolution formula, as one can just integrate the ODE twice, an advantage of our mean variant of the Sobolev metric. This saves one from having to compute the convolution directly, and hence a reduction in computational cost from quadratic (or quasi-linear with an FFT) to linear in $O$ given the $H^0$ gradient. The Sobolev gradient can be computed as

$$g(o, i, h, w) = g(0, i, h, w) + o\frac{\partial g}{\partial o}(0, i, h, w) - \frac{1}{\lambda} \int_0^o (o - \tilde{o})(f(oO, i, h, w) - \bar{f}(i, h, w)) \, \mathrm{d}\tilde{o} \quad (10)$$

$$\frac{\partial g}{\partial o}(0, i, h, w) = -\frac{1}{\lambda} \int_0^1 o(f(oO, i, h, w) - \bar{f}(i, h, w)) \, \mathrm{d}o \quad (11)$$

$$g(0, i, h, w) = \int_0^1 K(o) f(oO, i, h, w) \, \mathrm{d}o, \quad o \in [0, 1] \quad (12)$$

where $g = \nabla_{\tilde{H}^1} L(X)$ and $f = \nabla_{H^0} L(X)$. These are just three integrals that can be computed in linear complexity with respect to $O$. The gradient flows under these metrics are given by

$$\dot{X}_t = -\nabla L(X_t), \quad (13)$$

where $t$ denotes the artificial time variable, $\dot{X}$ is the time derivative of the parameter tensor, and $\nabla$ denotes the gradient with respect to the desired metric.

## 2.4 Properties of Channel Directed Gradient Flows

**Correlation in the Weight Tensor**: By the convolution formula, the Sobolev gradient is a smoothing of the $H^0$ gradient. Noting that the gradient flow (13) integrates (smooth) gradients over time, the final tensor will exhibit correlation in the output direction as it sums smooth (correlated) gradients in the output direction and the initialization, which is typically chosen to be decorrelated noise.

**Coarse-to-Fine Evolution and Removal of Some Local Minima**: Sobolev gradient flows evolve according to coarse-scale perturbations before moving to finer scale perturbations Sundaramoorthi et al. (2008). This avoids being trapped in local minima due to fine-scale structures. Also, since Sobolev balls can fit in any $\mathbb{L}^2$ ball but not vice-versa, the loss landscape changes (i.e., topologically in the continuum) and some local minima (in $\mathbb{L}^2$) may cease to exist numerically. As wide local minima generalize well Chaudhari et al. (2019), the numerical removal of local minima due to fine structures (e.g., sharp minima) may encourage convergence to wide minima and hence generalize better than SGD. The correlated nature of the Sobolev (and re-weighted $H^0$) gradient makes it difficult to lock into sharp local minima.

```
def reweighted_H0_grad(grad=param.grad.data, lambda):
    #grad: L2 gradient; lambda>0 weights translation of L2 grad
    grad += lambda*torch.mean(grad,0,True).repeat(grad.size(0),1,1,1)
    return grad
```

Figure 3: Pytorch code to compute the re-weighted $H^0$ ($H^0_\lambda$) gradient from the $H^0$ gradient.

## 3 APPLICATION TO SGD AND IMPLEMENTATION

To apply re-weighted $H^0$ and Sobolev channel-directed gradients to optimizing CNNs based on SGD or its variants, we discretize the gradient flow (13) according to forward Euler. We approximate the standard $H^0$ gradient of the loss, $\nabla_{H^0} L(X)$, using a mini-batch, as is standard. We then use this approximation of the $H^0$ gradient to approximate the $\tilde{H}^1$ gradient, $\nabla_{\tilde{H}^1} L(X)$, by discretizing (10)-(12) using a standard Riemann sum. Note that (10) can be computed for each $o$, the output channel index of the tensor, with the cumulative sum (CUMSUM) operation, which is linear in cost, as are (11) and (12). We compute the Sobolev gradient for each convolutional layer parameter tensor independent of others. We use $\lambda = 1$ for $\tilde{H}^1$ gradient and add it to a scaled version (by a hyper parameter) of the $H^0$ gradient (as in Figure 2) to avoid over-smoothing. The re-weighted $H^0$ gradient is computed by using (8) from the $H^0$ stochastic gradient. Both our gradients require few additional lines of code; the code for re-weighted $H^0$ is shown in Figure 3 (see Appendix Figure 12 for $\tilde{H}^1$ code). Thus, our gradients replace the usual one, and other additions to SGD (e.g., momentum, Adam) can be used.

## 4 EXPERIMENTS

We test our methods on different baseline optimizers and tasks. Our intent is to show that any method can be improved just by switching to either of our gradients. We fix $\lambda = 1$ unless specified otherwise. Table 1 shows the settings for each experiment. Experiments are run on a single NVIDIA Titan Xp GPU except for GANs, which are run on a Tesla v100 GPU due to memory requirements.

Table 1: **Experimental settings.**

| Task | Dataset | Baseline | Network | Batch Size | Epochs | Initial LR |
|------|---------|----------|---------|------------|--------|------------|
| Image Classification | CIFAR-10 | SGD | ResNet-56 | 128,32,8 | 240 | 0.1 |
| | | | VGG-16 | 128,8,6 | 240 | 0.01 |
| | | ADAM | ResNet-56 | 128,32,8 | 200 | 1e-3 |
| | | LS | ResNet-56 | 128,32,8 | 240 | 0.1 |
| | MNIST | SGD | Two-layer Conv | 100 | 100 | 0.01 |
| Semantic Segmentation | PascalVOC | SGD | ResNet50 | 2 | 70 | 7e-3 |
| Image Generation (GAN) | CityScapes | SGD | SPADE | 2 | 100 | 1e-4,4e-4 |

**Image Classification:** We experiment on CIFAR-10 Krizhevsky et al. (2009). We test our gradients with both SGD and ADAM on ResNet-56 He et al. (2016a) and VGG-16 Simonyan & Zisserman (2014) following settings of Osher et al. (2018). For SGD, we set the initial learning rate to be 0.1 and 0.01 on ResNet-56 and VGG-16 respectively with momentum 0.9 and weight decay 5e-4. For ADAM, we set the initial learning rate to 0.01. We decrease the learning rate by a factor of 10 every 40 epochs as Osher et al. (2018). We run 25 independent trials on SGD and 10 on ADAM (due to lower variance of ADAM), and report the average.

In Figure 4, we show an example of training and test accuracy curves (batch size of 8) for baselines as well as Laplacian Smoothing (LS) Osher et al. (2018), which rasterizes before smoothing. We out-perform all methods. We also apply LS (without rasterization) to smooth the gradient in our output-channel directed fashion, which improves LS, but we still out-perform it.

In Figure 5 (left), we compare the histograms of test accuracy over multiple runs of ours and SGD. Our method achieves higher average test accuracy with reduced variance. To investigate the effect of different channel directions of smoothing, we apply our method as well as LS along different channel-directions. We compare approaches under two settings, which are smoothing gradients in all layers and smoothing gradients in only convolutional layers. Figure 5 (right) shows that our output-channel direction is preferred regardless of smoothing method used. This shows that the output channel smoothing is essential. Smoothing only convolutional layers in a rasterized order (as

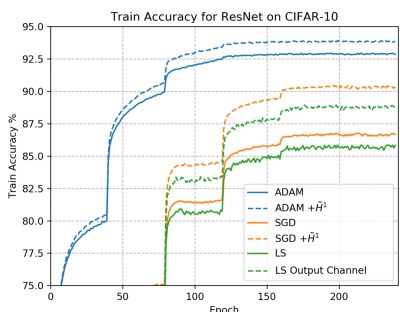
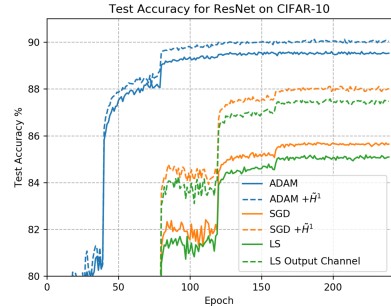

Figure 4: **Evolution of training and test accuracy on CIFAR-10: an example with batchsize = 8.** Our metric improves both training and test accuracy.

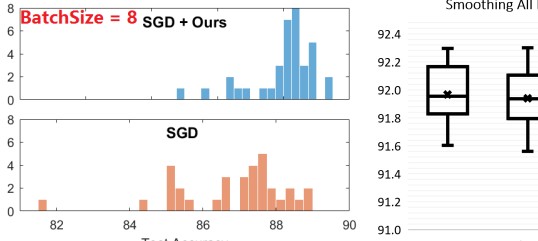
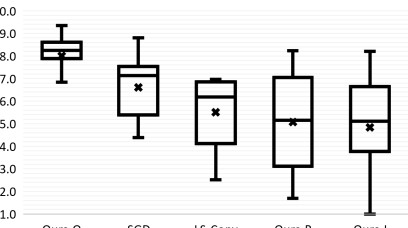

Figure 5: **Distribution of results on CIFAR-10.** *Left*: Histogram of test accuracy. Ours achieves higher average with significantly reduced variance. *Right*: Results from different methods. Best accuracy obtained from our proposed direction. *Ours: SGD+$\tilde{H}^1$; LS-ChanDir: LS applied in our proposed channel direction; Ours+O: output channel smoothing; Ours+R: parameters rasterized into a 1-D vector to perform smoothing; Ours+I: Input channel smoothing.*

in LS) performs worse than SGD, further showing importance of the output direction. Note LS makes up the loss in convolutional layer smoothing with smoothing in other layers.

Table 2 summarizes results (over 10-25 trials). Both our gradients improve over $H^0$. A greater advantage is achieved with small batch sizes is small as the stochastic gradient is noisy, and our method imposes regularity. Both gradients perform similarly, but $\tilde{H}^1$ performs better with ADAM.

Table 2: **Test accuracy on CIFAR-10.** Channel-directed gradients improve $H^0$ in all cases. Up to 11% of errors can be reduced. Results average 25 trials for SGD and 10 trials for ADAM.

| Architecture | ResNet-56 | | | VGG-16 | | | Architecture | ResNet-56 | | |
|---|---|---|---|---|---|---|---|---|---|---|
| Batch size | 128 | 32 | 8 | 128 | 8 | 6 | Batch size | 128 | 32 | 8 |
| SGD | 93.24 | 91.96 | 86.54 | 93.02 | 92.31 | 91.88 | ADAM | 91.20 | 91.04 | 89.53 |
| $+\tilde{H}^1$ | 93.39 | 92.27 | 87.99 | 93.26 | 92.77 | 92.25 | $+\tilde{H}^1$ | 91.42 | 91.13 | 90.02 |
| Error reduced% | 2.2% | 3.9% | 10.8% | 3.4% | 6.0% | 4.6% | Error reduced% | 2.5% | 1.0% | 4.7% |
| $+H_\lambda^0$ | 93.57 | 92.25 | 88.04 | 93.19 | 92.79 | 92.43 | $+H_\lambda^0$ | 91.20 | 91.06 | 89.70 |
| Error reduced% | 4.9% | 3.6% | 11.1% | 2.4% | 6.2% | 6.8% | Error reduced% | 0.0% | 0.2% | 1.6% |

**Effect of Smoothing Parameter**: We examine the effect of the smoothing parameter on MNIST Le-Cun & Cortes (2010) and Fashion-MNIST Xiao et al. (2017) by varying it from 0 to 20. We conduct training on the test set (10000 samples) and test on the training set (60000 samples) to make generalization more challenging. We use a 2-layer CNN with 50 and 100 $5 \times 5$ filters in each layer, respectively, and train with batch size 100. Figure 6 shows the accuracy at the 100th epoch (average over 5 trials). Note $\lambda = 0$ is SGD. Our methods are not sensitive to $\lambda$ and improve SGD for any $\lambda$.

**Semantic Segmentation:** The experiments are conducted on PascalVOC Everingham et al. (2015) using the popular UNet segmentation network Ronneberger et al. (2015) with ResNet-50 as the encoder (https://github.com/nyoki-mtl/pytorch-segmentation). We use initial learning rate 7e-3 and batch size 2 (to fit on Titan Xp memory), and average results over 3 trials. Figure 7 shows results. Both our gradients improve segmentation accuracy by ~8% over SGD on the test set. We reduced the generalization gap from 0.163 to 0.151 (by 7.4%) and 0.150 (by 8.0%) for $\tilde{H}^1$ and $H_\lambda^0$, respectively.

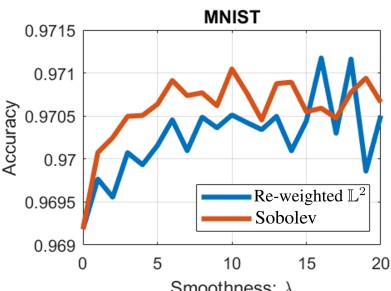 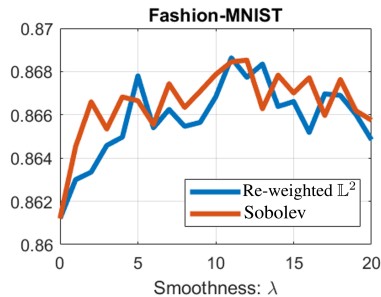

Figure 6: **Results on MNIST and Fashion-MNIST with different choice of smoothness.** Our methods improve classification accuracy over SGD (i.e., $\lambda = 0$) for a wide range of smoothness.

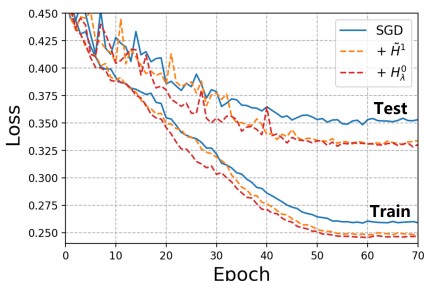 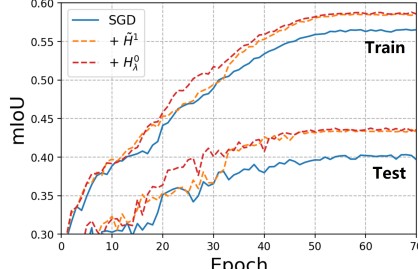

Figure 7: **Semantic Segmentation Results on PascalVOC.** Sobolev $\tilde{H}^1$ and re-weighted $H^0$ ($H^0_\lambda$) improve segmentation accuracy by 8.5% and 7.8% respectively relative to SGD.

**GAN Image Generation:** We test on semantic labels to image conversion. We perform the experiments on the current state-of-the-art model SPADE Park et al. (2019). We test on CityScapes Cordts et al. (2016) dataset and results are evaluated by FID Heusel et al. (2017) score (lower is better). Learning rates are $1e-4$ and $4e-4$ for the generator and discriminator, respectively. We compare to SGD with momentum 0.9 and weight decay 5e-4. All models are trained with batch size 2 (to fit on Tesla v100 memory). For each optimizer, we summarize the results of 24 different trained models. Table 3 provides results. Our methods achieve better average FID score with less variance.

| Method | FID |
|---|---|
| SGD | $65.77 \pm 11.94$ |
| $+\tilde{H}^1$ | $60.17 \pm 6.15$ |
| $+H^0_\lambda$ | $57.99 \pm 5.01$ |

Table 3: **Results on the image generation task.** Our methods achieve better result with reduced variance due to regularity imposed during training.

**Speed:** With PyTorch, re-weighted $H^0$ adds negligible overhead. Currently, $\tilde{H}^1$ increases training time on CIFAR-10 by 50% with batch size 128. 70% of this overhead is due to using tensor transpose and saving/loading, which is required due to limitations of Pytorch library. This can be eliminated by implementing our own Pytorch function in C++; in this case, $\tilde{H}^1$ would add a 15% overhead.

## 5 CONCLUSION

Using stochastic gradients that promote correlation (and smoothness) in the output-channel dimension of CNN network tensors is effective in improving accuracy of SGD and its variants. We reformulated the gradient (without changing the loss) by changing the underlying Riemannian geometry on the tensor space using two different metrics. In the continuum, Sobolev changes the topology of the loss landscape (possibly removing fine-scale local minima), and so has better theoretical properties. Both the channel-directed re-weighted $H^0$ and $\tilde{H}^1$ gave accuracy boosts, with $\tilde{H}^1$ performing better with ADAM. Regularity in other tensor dimensions is not effective in improving accuracy. Both channel-directed gradients have the same (linear) computational complexity and not much cost over SGD (re-weighted $H^0$ is faster), and the code is simple.

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

# A    ADDITIONAL ANALYSIS OF EVOLUTION OF CHANNEL-DIRECTED OPTIMIZATION

Figure 8 and Figure 9 present the evolution of training and test accuracy of ADAM and SGD with different batch sizes. Using channel-directed gradients ($\tilde{H}^1$ in this experiment) for SGD or ADAM improves test accuracy for any batch size. More prominent performance gains are seen for smaller batch sizes. This is due to that the stochastic gradient is typically more noisy when the batch size is small, and our proposed channel-directed metrics implicitly encode smoothness.

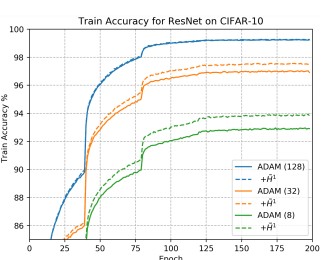 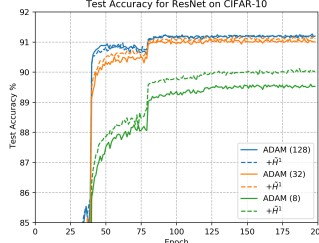

Figure 8: **Training and test accuracy on CIFAR-10 with ADAM.**

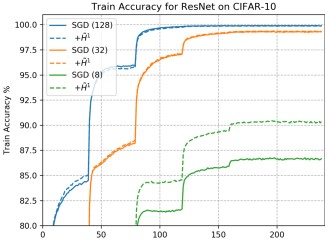 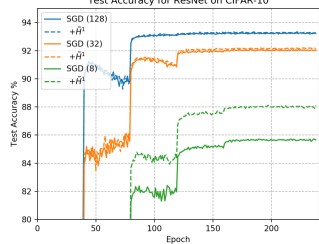

Figure 9: **Training and test accuracy on CIFAR-10 with SGD.**

# B    ADDITIONAL EXPERIMENTAL VERIFICATION OF OUTPUT-CHANNEL DIRECTION

To investigate the effect of different channel directions of smoothing, we apply our method as well as LS along different channel-directions. Figure 10 shows that our output-channel direction is preferred regardless of different smoothing approaches.

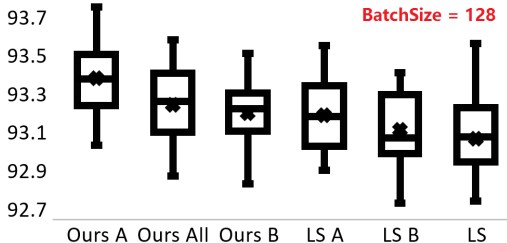

Figure 10: **Channel-Directed Smoothing Leads to Better Performance.** Best accuracy obtained from our proposed direction. *A: Output-Channel Directed; B: Input-Channel Directed; All: parameters rasterized into a 1-D vector to perform smoothing; Ours: re-weighted $\mathbb{L}^2$.*

## C    REGULARITY OF TRAINED CONVOLUTIONAL LAYERS

We show that the final weight tensor at convergence in our methods have correlation in the output channel dimension in Figure 11, as should be the case as the tensor is composed of a component that is smooth. To show this, we plot the correlation between filters in the weight tensors as a function of the distance in the output channel dimension. This is done over multiple tensor layers in ResNet-56 and over multiple trials of optimization on CIFAR-10. We also show the correlation of filters in the input channel direction. As can be seen, all optimization methods produce tensors that exhibit correlation (in additional smoothness for Sobolev) in the output channel direction while no (or much less) correlation in the input direction. Notice that our methods increase the amount of regularity compared to SGD as it imposes this in optimization.

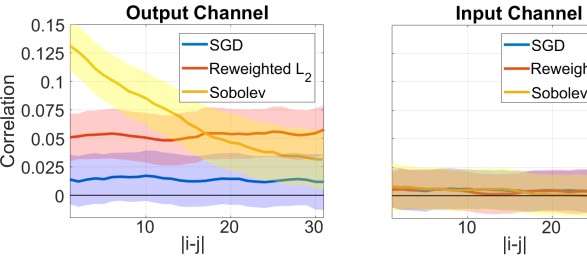

Figure 11: **Correlation of Final Tensor.** Correlation between weights within different channel directions in CIFAR trained ResNet 56 conv layers (over 10 trials). $|i - j|$ is distance between weight locations in tensor for correlation computation. Sobolev/re-weighted $H^0$ show strong correlation in output direction, but not input. SGD shows correlation in output direction.

## D    DETAILED DERIVATIONS FOR SECTION 2.2

We first derive the re-weighted $\mathbb{L}^2$ gradient under $H_\lambda^0$ metric following the same notations from the paper. Consider $f \triangleq \nabla_{H^0} L(X)$ the standard $\mathbb{L}^2$ gradient, and we want to solve for $g \triangleq \nabla_{H_\lambda^0} L(X)$. By (4) and (7) we have

$$\langle f, k \rangle_{H^0} = \langle g, k \rangle_{H_\lambda^0} \tag{14}$$

$$= \langle \bar{g}, \bar{k} \rangle_{H^0} + \lambda \langle g - \bar{g}, k - \bar{k} \rangle_{H^0}. \tag{15}$$

Note the fact that $\langle \bar{g}, k - \bar{k} \rangle_{H^0} = 0$ holds for all $k$. This is because $\int \bar{g}(k - \bar{k}) \, \mathrm{d}o = \bar{g} \int (k - \bar{k}) \, \mathrm{d}o$ and $\int (k - \bar{k}) \, \mathrm{d}o = 0$ since $k - \bar{k}$ is zero-mean. In this way, $\bar{k}$ and $k - \bar{k}$ become a set of orthogonal basis.

After decomposing $f$ and $k$ into

$$f = \bar{f} + (f - \bar{f}), \quad k = \bar{k} + (k - \bar{k}), \tag{16}$$

by simple algebra we have

$$\bar{f} = \bar{g}, \quad f - \bar{f} = \lambda(g - \bar{g}), \tag{17}$$

which leads to the result of (8).

We then derive the Sobolev gradient under $H^1$ metric, following similar computations in Sundaramoorthi et al. (2007). Consider $\nabla_{H^1} L(X)$ the Sobolev gradient under $H^1$ metric. By (6) and (7) we have

$$\langle \nabla_{H^0} L(X), k \rangle_{H^0} = \langle \nabla_{H^1} L(X), k \rangle_{H^1} \tag{18}$$

$$= \frac{1}{O} \langle k, \nabla_{H^1} L(X) \rangle_{H^0} + \lambda O \left\langle \frac{\partial k}{\partial o}, \frac{\partial \nabla_{H^1} L(X)}{\partial o} \right\rangle_{H^0}. \tag{19}$$

Integrating by parts and considering the periodic boundary conditions, we have

$$\langle \nabla_{H^0} L(X), k \rangle_{H^0} = \left\langle \nabla_{H^1} L(X) - \lambda O^2 \frac{\partial^2}{\partial o^2} \nabla_{H^1} L(X), k \right\rangle_{H^0}. \tag{20}$$

Since $k$ can be any perturbation, by uniqueness, we have

$$\nabla_{H^0} L(X) = \nabla_{H^1} L(X) - \lambda O^2 \frac{\partial^2}{\partial o^2} \nabla_{H^1} L(X) \tag{21}$$

which is (8). Similarly, for $\tilde{H}^1$ metric, we have

$$\nabla_{H^0} L(X) = \overline{\nabla_{\tilde{H}^1} L(X)} - \lambda O^2 \frac{\partial^2}{\partial o^2} \nabla_{\tilde{H}^1} L(X). \tag{22}$$

First observe that by computed the output-channel directed average of the both sides of the above equation, we see that $\overline{\nabla_{\tilde{H}^1} L(X)} = \overline{\nabla_{H^0} L(X)}$, i.e., the average values are same. One may integrate (22) twice to solve for the $\tilde{H}^1$ gradient. For simplicity, let $f$ be the $\mathbb{L}^2$ gradient and $g$ be the $\tilde{H}^1$ gradient. Integrating twice yields

$$g(o, i, h, w) = g(0, i, h, w) + \int_0^o \frac{\partial g}{\partial o}(0, i, h, w)\, d\tilde{o} - \frac{1}{\lambda} \int_0^o \int_0^{\hat{o}} (f(\tilde{o}O, i, h, w) - \bar{f}(i, h, w))\, d\tilde{o}\, d\hat{o} \tag{23}$$

$$= g(0, i, h, w) + \int_0^o \frac{\partial g}{\partial o}(0, i, h, w)\, d\tilde{o} - \frac{1}{\lambda} \int_0^o \int_{\tilde{o}}^o (f(\tilde{o}O, i, h, w) - \bar{f}(i, h, w))\, d\hat{o}\, d\tilde{o} \tag{24}$$

$$= g(0, i, h, w) + o\frac{\partial g}{\partial o}(0, i, h, w) - \frac{1}{\lambda} \int_0^o (o - \tilde{o})(f(\tilde{o}O, i, h, w) - \bar{f}(i, h, w))\, d\tilde{o}. \tag{25}$$

Note that here we perform normalization by scaling to the channel direction by letting $o \in [0, 1]$. With boundary conditions $g(0) = g(1)$, $\frac{\partial g}{\partial o}(0) = \frac{\partial g}{\partial o}(1)$ and $\bar{f} = \bar{g}$, we have

$$\frac{\partial g}{\partial o}(0, i, h, w) = -\frac{1}{\lambda} \int_0^1 o(f(oO, i, h, w) - \bar{f}(i, h, w))\, do. \tag{26}$$

For simplicity, we eliminate $i, h, w$ and $O$ in the following derivations. We have

$$g(0) = g(o) - o\frac{\partial g}{\partial o}(0) + \frac{1}{\lambda} \int_0^o (o - \tilde{o})(f(\tilde{o}) - \bar{f})\, d\tilde{o} \tag{27}$$

$$= g(o) + o\frac{1}{\lambda} \int_0^1 o(f(o) - \bar{f})\, do + \frac{1}{\lambda} \int_0^o (o - \tilde{o})(f(\tilde{o}) - \bar{f})\, d\tilde{o}. \tag{28}$$

Noting $\int_0^1 g(0)\, do = g(0)$ and $\bar{f} = \int_0^1 f(o)\, do$, we integrate both sides over the entire interval $[0, 1]$.

$$g(0) = \bar{g} + \frac{1}{\lambda} \int_0^1 o\, do \cdot \int_0^1 o(f(o) - \bar{f})\, do + \frac{1}{\lambda} \int_0^1 \int_0^o (o - \tilde{o})(f(\tilde{o}) - \bar{f})\, d\tilde{o}\, do \tag{29}$$

$$= \bar{f} + \frac{1}{2\lambda} \int_0^1 of(o)\, do - \frac{1}{4\lambda}\bar{f} + \frac{1}{\lambda}(\int_0^1 \int_0^o (o - \tilde{o})f(\tilde{o})\, d\tilde{o}\, do + \bar{f} \int_0^1 \int_0^o (o - \tilde{o})\, d\tilde{o}\, do \tag{30}$$

$$= (1 - \frac{1}{4\lambda} - \frac{1}{6\lambda})\bar{f} + \frac{1}{2\lambda} \int_0^1 of(o)\, do + \frac{1}{\lambda} \int_0^1 \int_{\tilde{o}}^1 (o - \tilde{o})f(\tilde{o})\, do\, d\tilde{o} \tag{31}$$

$$= (1 - \frac{5}{12\lambda}) \int_0^1 f(o)\, do + \frac{1}{2\lambda} \int_0^1 of(o)\, do + \frac{1}{\lambda} \int_0^1 (\frac{1}{2} + \frac{\tilde{o}^2}{2} - \tilde{o})f(\tilde{o})\, d\tilde{o} \tag{32}$$

$$= \int_0^1 (1 + \frac{o^2 - o + 1/6}{2\lambda})f(o)\, do. \tag{33}$$

This gives (12) in the main paper.

## E    Code for Sobolev Gradient

The Pytorch code to compute the Sobolev gradient is provided in Figure 12. In theory, the 'cumsum' operation should be the main part of the code with largest computational cost. However, in order to match with standard Pytorch library, tensor operations including 'permute', 'repeat' and 'unsqueeze' are currently required. These operations contribute to over 70% of computational overhead, and can be avoided by if the computation were done using C++.

```
def Sobolev_grad(grad=param.grad.data,lambda):
    #grad: L2 gradient; lambda>0
    L = grad.size(0)
    s = torch.arange(L,dtype=torch.float32).cuda()
    tmp_mean = torch.mean(grad,0, True).repeat(L,1,1,1)
    tmp_diff = (grad − tmp_mean).permute(1,2,3,0)
    gp_0 = torch.matmul(tmp_diff,s)/(−lambda*L**3)
    gp_0 = gp_0.unsqueeze_(3).repeat(1,1,1,L)
    s = s.unsqueeze_(0).unsqueeze_(0).unsqueeze_(0).repeat
        (tmp_diff.size(0),tmp_diff.size(1),tmp_diff.size(2),1)
    # Sobolev gradient computation
    tmp2 = s*gp_0−(s*torch.cumsum(tmp_diff,dim=3)
        − torch.cumsum(s*tmp_diff,dim=3))/(lambda*L**2)
    grad = lambda*(tmp2.permute(3,0,1,2) + tmp_mean)
    return grad
```

Figure 12: **Pytorch code to compute the Sobolev ($\tilde{H}^1$) gradient from the $H^0$ gradient.** The 'permute', 'repeat' and 'unsqueeze' operations are due to standard library limitations, and can be avoided by further code optimization (e.g., writing the function in C++/Cuda that Pytorch calls).

## F  FURTHER ANALYSIS OF CORRELATION IN CONVOLUTIONAL LAYERS

Existing analysis on regularity in CNNs (Mao et al. (2017)) focus on filter level and kernel level regularities in pruning. To the best of our knowledge, the channel-directed regularity proposed by our paper has not been investigated nor has the regularity been used in optimization.

We show below that the output-direction correlation is not due to randomness in the network or due to particular weight transformations that leave the output behavior of the CNN fixed. The reason for the output direction correlation is unknown to us, but a direction for future investigation.

**Could it be due to random noise?** No. Figure 13 shows the histogram of correlation of a representative convolutional layer from ImageNet pretrained DenseNet. Note that the input channel correlation is distributed as zero-mean Gaussian, which is likely to be due to random noise. In contrast, the output channel correlation (our proposed channel direction) shows positive correlation. There are also outliers (points near 0.3) corresponding to channels with mean value far away from zero (see vertical lines in Figure 1). This shows that the neural network prefers regularities in the output channel direction.

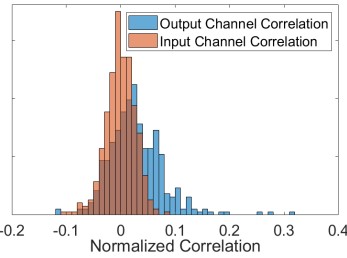

Figure 13: **Histogram of correlation of a representative tensor.** While the input channel correlation is distributed as zero-mean Gaussian, the output channel shows positive correlation and sparsity. The neural network prefers regularities in the output channel direction.

**Could it be due to scaling?** No. In modern CNNs, scaling the affine factor in BatchNorm layer could create such structure in the following convolutional layer, without affecting the output of the neural network. Note that in Figure 1, we use correlation that is invariant to re-scalings so this is not the case. We also investigate this further. Figure 14 presents the scatter plot of mean of tensors in the output direction (larger mean corresponds to stronger correlation) and standard deviation of output channels. If scalings are contributing to the regularity, there would be a positive correlation between the mean and standard deviation, as a scaling amplifies both. The plot does NOT show a positive

correlation between mean and standard deviation in this channel direction, which means the structure is not due to simply scaling up all weights within particular channels (producing the same CNN).

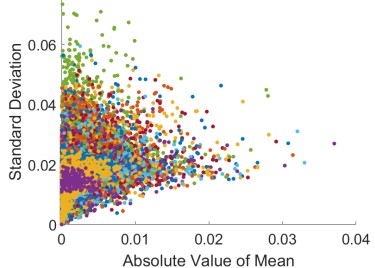

Figure 14: **Scatter plot of mean and standard deviation of output channels.** There is no positive correlation between channel mean and standard deviation, showing that the structure in output channel direction is not due to scaling. *Each color corresponds to a layer.*

