# OpenReview forum: "Channel-Directed Gradients for Optimization of Convolutional Neural Networks"
_ICLR.cc/2021/Conference — Reject_

### Official Review · AnonReviewer4 · 2020-10-29
**Prof.**

**Rating:** 6
**Confidence:** 3

**Review:**

In this paper propose a stochastic optimization method for CNNs, which can obtain an improvement in terms of generation error. The proposed method is used to compute the gradient with respect to output-channel directed re-weighted matrices or Sobolev metrics, and some anabasis is also provided. Some experimental results show that the proposed method can be improved in generalization error.

1. It is not clear that what's the relationship between input channel, output channel correlations and Sobolev gradients.
2. What's the key difference between Laplacian smoothing and Sobolev gradients?
3. The relationship and difference between channel-directed gradients and Sobolev gradients should be analyzed.
4. The experimental analysis is less convincing. The authors should compare the proposed algorithm with sophisticated methods such as manifold optimization algorithms.

---

> ### Author Response · Authors · 2020-11-13
> **Clarifications of Sobolev, channel-directed, Laplacian smoothing, and other manifold optimization**
>
> 1) It is not clear that what's the relationship between input channel, output channel correlations and Sobolev gradients.
>
> -- Sobolev gradients are general gradients that impose smoothness/correlation in the gradient that induces correlation in the parameter tenssor. In this paper we define our channel-directed Sobolev gradients to further impose smoothness/correlation only in the output channel direction that leads to such correlation in the output direction of the tensor (see Sect 2.4 first paragraph).
>
>
> 2) What's the key difference between Laplacian smoothing and Sobolev gradients?
>
> -- The standard form of the Sobolev metric gives a Sobolev gradient that is effectively computed using Laplacian smoothing. The Sobolev metric that we propose is based on the mean value of the tensor, and leads to gradient computations which have less computational complexity than Laplacian smoothing (O(n) vs O(nlog(n)), page 5, 'This saves one ...'). The previous paper that introduced Laplacian smoothing, however didn't look into channel directed versions.
>
> The framework that we are presenting illustrates the process of how prior knowledge in the structure of the tensor can be imposed in the metric and hence the gradient. It illustrates how in general one can start with a prior, encode it in a metric and impose in the optimization procedure. Although computationally, the Sobolev gradient reduces down to Laplacian smoothing with a particular form of Sobolev metric, the fact that we formulate this as changing the underlying Riemannian geometry to derive gradients more general than Laplacian smoothing. We hope this framework will motivate new research into new metrics.
>
>
> 3) The relationship and difference between channel-directed gradients and Sobolev gradients should be analyzed.
>
> -- Sobolev gradients are general gradients with smoothness; channel-directed Sobolev is a type of Sobolev gradient with smoothness in particular directions of the tensor.  Channel-directed reweighed L2 is a version of the L2 with certain channel direction components more heavily weighted. In practice, both versions of channel-directed perform similarly but channel-directed Sobolev performs better with ADAM.
>
>
> 4) The authors should compare the proposed algorithm with sophisticated methods such as manifold optimization algorithms.
>
> -- Please see response 2) to Anonymous Reviewer 5.

---

### Official Review · AnonReviewer1 · 2020-10-29
**weak baselines, needs more work**

**Rating:** 4
**Confidence:** 3

**Review:**

The paper proposes a modification of the gradients of objective function wrt model parameters by changing the underling metric on the manifold of model parameters.  It proposes two metrics, a reweighed L2 metric, and a specific Sobolev metric, and presents how to compute modified gradients in a compute efficient manner.  Motivated by empirical observation that parameters of trained CNN tensors tend to be correlated along output dimension, the paper focuses on metrics that promote such correlation.

The approach of considering new metrics is appealing in that it allows keeping original loss function, model parameterization, and optimization methodology; it modifies the gradients in potentially beneficial ways leading to better optimization.  The formulation presented is mathematically rigorous.

The biggest criticism I have is that the baseline systems presented in the paper appear very far behind the best known results. For instance,
    * State of the art accuracy on CIFAR-10 is over 99% but the models presented in the paper achieve below 90%. This is a very large gap.
    * For PascalVOC the mean IoU is in high 80s, not sure why the paper cites numbers near 40. I’ve re-read the paper a few times to find out what am I missing but can’t tell why the numbers are so different.
With baselines like that it becomes very hard to assess the real value of the proposed approach.

Another weak aspect of the paper is the choice of output channel dimension for smoothing.  It is empirically motivated based on the correlations in parameters along output dimension.  At the same time results also show that other dimensions such as input channel dimension lead to accuracy degradation.  However, since the method allows for selecting any subset of variables to smooth over, could there be more principled approaches for identifying optimal subset?  Could the optimal metric be somehow learnt?

For aforementioned reasons I feel the paper needs more work prior to publication.

---

> ### Author Response · Authors · 2020-11-13
> **Baselines are current best optimizers; we test on the most widely used nets**
>
> 1) The baseline systems presented in the paper appear very far behind the best known results.
>
> -- Our goal is demonstrate the performance of the optimizer, as our contribution is to introduce a new optimization technique.  Neural networks can be improved by better architectures and optimizers, but our goal as a research paper is to explore the optimization axis. Thus, our experiments fix the architectures to be widely used ones in practice (e.g., VGG, ResNet), and show improvement in the optimization over the baseline state-of-the-art optimizers (SGD & ADAM).  We show how merely changing the underlying metric can lead to a different gradient with nearly no additional cost. For PascalVOC we train on small batch size (page 7) due to memory limitations so the result could be lower, but again, both of our methods significantly outperform SGD.
>
>
>
> 2) Could there be more principled approaches for identifying optimal subset? Could the optimal metric be somehow learnt?
>
> -- We agree that this is an interesting point and an interesting future direction of research, which is not within the scope of this paper.

---

### Official Review · AnonReviewer3 · 2020-11-01
**An intersting paper on CNN optimization**

**Rating:** 6
**Confidence:** 1

**Review:**

This paper proposes an optimization method for convolutional neural networks that can be used to improve existing gradient-based optimization in terms of generalization error.  The method computes the gradient of the loss function with respect to output-channel directed re-weighted $H^0$ or Sobolev metrics.

The paper has the following merits:
+ The paper shows that defining the gradients along the output channel direction leads to a performance boost, while other directions can be detrimental. I wonder whether the method can be used for other networks beyond CNN?
+ The continuum theory of gradients, its discretization, and application to deep networks are provided.
+ The effectiveness of the proposed method is demonstrated by experiments on benchmark datasets, several networks, and baseline optimizers.
+ The method requires only simple processing of existing stochastic gradients, can be used in conjunction with any optimizer, and has only a linear overhead compared to computation of the stochastic gradient.

The paper is well structured and well written; while I was not able to confirm every single proof and algorithm, I am confident that the topics are presented in a sound way and would make an interesting topic at the conference.

-------------------------------------------------------
I have read the response, and the rating is not changed.

---

> ### Author Response · Authors · 2020-11-13
> **beyond CNNs**
>
> 1) I wonder whether the method can be used for other networks beyond CNN?
>
> -- There could be similar structure in parameter weights in networks more general than CNNs, which could be an interesting direction for future work.
>
> Thanks for the positive review.

---

### Official Review · AnonReviewer5 · 2020-11-07
**The proposed method should be further explained, and additional analyses should be provided.**

**Rating:** 5
**Confidence:** 5

**Review:**


The paper proposes a method to compute the gradient of the loss function with respect to output-channel directed re-weighted H0 or Sobolev metrics, which has the effect of smoothing components of the gradient across a certain direction of the parameter tensor. The proposed method was used to train CNNs with gradient based optimisation methods such as SGD and Adam.

In general, the proposed method and motivation are interesting. Experimental results show that the proposed method improves baseline accuracy for small batch sizes.

However, there are some major issues with the proposed method and analyses:

- In the proposed method, Sobolev gradients are formulated by considering the space of parameter tensors as a Riemannian manifold, and choosing the Sobolev metric on the tangent space. That is, the method is proposed as a new optimisation method on Riemannian manifolds of parameters. Therefore, the following issues should be resolved by additional analyses and explanations:

— How do you identify the Riemannian manifold of these tensors?
— How do you assure that tensors remain on the manifold during training of networks?
— Please compare the proposed method with other Riemannian optimisation methods applied to train CNNs in the experimental analyses.

- Please provide variance of accuracy for all the experiments, esp. for classification tasks to analyse statistical significance of results, since mean accuracy values of baseline and the proposed methods are very close to each other.

- Parameter correlations for CNNs trained on Imagenet were analysed in Figure 1. However, additional analyses on employment of the proposed method for these CNNs were not given in the experimental analyses. Please apply the proposed method on larger benchmark datasets such as Imagenet, and provide these results as well. These results may be also used to examine the claim proposed in Figure 1.

- The proposed code snippets were useful to see how of the methods are implemented using PyTorch. However, we implemented similar experiments using these code snippets, but could not achieve similar results (indeed, obtained worse accuracy wrt baseline in some cases). A reason of this observation may be difference in implementation of other parts. Therefore, it would be useful and more helpful if you could provide the complete, esp. for experiments on image classification using Cifar 10 and semantic segmentation using Pascal VOC.


After the rebuttal:

I checked comments of other reviewers and response of authors.

First, thank you for the detailed response. Since some of my questions and concerns are partially addressed, I improve the overall rating.

However, there are still parts which should be improved:

- Regarding manifolds: There are some statements which should be clarified and appropriately analyzed in the paper. For instance, it is stated that "The manifold of parameter tensors is a linear space, and so any linear combination of the parameter set with a tangent vector (another tensor) will remain on the manifold. We are changing the metric on the tangent space from the ordinary Euclidean metric to our channel directed (Sobolev, and re-weighted L2) metrics, which effectively changes the lengths of paths on this manifold of parameter tensors, making it a non-trivial Riemannian manifold."
-- This statement claims that the "structure" of the manifold of tensors changes by changing the metric on tangent space by changing geodesic or paths on manifolds. However, it is not clear how the geodesic or in general, geometry of the manifold changes by just changing the metric on the tangent space (while, the tangent space can change depending on change of the manifold).
-- Then, it is claimed that this leads to a non-trivial Riemannian manifold (a nonlinear space), while it is also claimed that the manifold of parameter tensors is a linear space (why is it a linear space?).

- Regarding experimental results: Thank you for the updated results. However, these results show that the accuracy of the proposed method is pretty close to the baseline. To show superiority of the proposed method, the experiments should be extended using larger benchmark datasets.

-- There are Riemannian optimization method that encodes this channel-directed structure, but there are various Riemannian optimization methods which claim to improve accuracy of models in various tasks such as the following:

S. Kumar Roy, M. Harandi, R. Hartley and R. Nock, Siamese Networks: The Tale of Two Manifolds. (oral), Int. Conference on Computer Vision (ICCV), Seoul, 2019.

Lei Huang, Xianglong Liu, Bo Lang, Admas Wei Yu, Bo Li, Orthogonal Weight Normalization: Solution to Optimization over Multiple Dependent Stiefel Manifolds in Deep Neural Networks, AAAI 2018 (Oral)

Therefore, the proposed methods should be compared with these methods in the analyses as well.

---

> ### Author Response · Authors · 2020-11-13
> **Theoretical questions clarified; new statistical significance results; correlations of trained nets in appendix**
>
> 1) How do you identify the Riemannian manifold of these tensors? How do you assure that tensors remain on the manifold during training of networks?
>
> -- The manifold of parameter tensors is a linear space, and so any linear combination of the parameter set with a tangent vector (another tensor) will remain on the manifold. We are changing the metric on the tangent space from the ordinary Euclidean metric to our channel directed (Sobolev, and re-weighted L2) metrics, which effectively changes the lengths of paths on this manifold of parameter tensors, making it a non-trivial Riemannian manifold.
>
>
> 2)   Please compare the proposed method with other Riemannian optimisation methods applied to train CNNs in the experimental analyses.
>
> -- Our intention in this paper is to show how prior structure in the parameter tensors can be favored through changing the underlying Riemannian metric on the manifold of tensors, which effectively gives a gradient with that structure encoded.  We are not aware of any other Riemannian optimization method that encodes this channel-directed structure. While there are optimization methods leveraging Riemannian Geometry (e.g. natural gradient methods), they do not have the same purpose as encoding this prior structure (e.g., natural gradients are to obtain re-parameterization invariance).  We should note that Laplacian Smoothing (which we compared to) can also be interpreted as a Riemannian optimization (though not mentioned in that work), and we out-perform it, though it encodes a different prior structure.
>
>
> 3) Provide variance of accuracy.
>
> -- Figure 5 and Figure 10 do show the statistical significance by box plots. Note that accuracy reported on Cifar are the average of over 25 independent trials for SGD and 10 trials for ADAM (page 6, 'We run 25 ...').  To better demonstrate the statistical significance of the improvement, we performed a t-test on the Cifar results. The results including variance and statistical significance (p<0.05) are listed below.  We will add to the updated version of the paper as suggested by the reviewer.
>
> | Architecture     |ResNet-56|ResNet-56|ResNet-56|VGG-16|VGG-16|VGG-16| Architecture|ResNet-56|ResNet-56|ResNet-56|
> |------------------|---------|-------|-------|-------|-------|-------|----------------|-------|-------|-------|
> | Batch Size       | 128     | 32    | 8     | 128   | 8     | 6     | Batch Size     | 128   | 32    | 8     |
> | SGD              | 93.24   | 91.96 | 86.54 | 93.02 | 92.31 | 91.88 | SGD            | 91.20 | 91.04 | 89.53 |
> | Variance         | 0.037   | 0.051 | 4.540 | 0.027 | 0.280 | 0.140 | Variance       | 0.020 | 0.059 | 0.149 |
> | +$\tilde{H^1}$   | 93.39*  | 92.27*| 87.99*| 93.26*| 92.77*| 92.25*|+$\tilde{H^1}$  | 91.42*| 91.13 | 90.02*|
> | Variance         | 0.033   | 0.060 | 1.385 | 0.018 | 0.210 | 0.203 | Variance       | 0.037 | 0.051 | 0.142 |
> | +$H^0_{\lambda}$ | 93.57*  | 92.25*| 88.04*| 93.19*| 92.79*| 92.43*|+$H^0_{\lambda}$| 91.20 | 91.06 | 89.70 |
> | Variance         | 0.034   | 0.038 | 0.692 | 0.023 | 0.088 | 0.154 | Variance       | 0.006 | 0.088 | 0.032 |
>
> *: the improvement is statistically significant (p<0.05).
>
>
> 4) Parameter correlations for CNNs ... These results may be also used to examine the claim proposed in Figure 1.
>
> -- We do provide numerical analysis of correlations on trained convolutional layers by our proposed method in Appendix C (page 12) on CIFAR. Both reweighted L2 and Sobolev optimized networks show stronger correlation in the output direction than SGD trained networks. Note SGD does still show correlation, but smaller than networks trained with our optimizers.
>
>
> 5) Implementation details; reproducibility.
>
> -- We will make the code publicly available. The readers are free to verify the results. In the experiments, we used exactly the same environment and settings as LS (Osher et al. 2018). Detailed training parameters including learning rate schedule, momentum and weight decay are provided in Section 4 and Table 1. Due to stochasticity there could be outliers in individual trials but all results we report are the average over multiple trials (page 6, 'We run 25 ...').

---

### Official Review · AnonReviewer2 · 2020-11-07
**Channel-Directed Gradients for Optimization of Convolutional Neural Networks**

**Rating:** 6
**Confidence:** 2

**Review:**

This paper follows a very active line of research on gradient-based optimization methods for convolutional neural networks. In particular, this work proposes a method that can be applied as an extension to popular optimization methods such as SGD and Adam. The main idea is to introduce a modification of the tensor space in order to encourage correlations in the output-channel dimension. In particular, they propose two  channel re-weighting strategies: H^0 and H^1.  The theoretical foundation behind the paper seems to be sound and the overall writing is clear. Results indicate that, indeed, the proposed modifications help to improve the performance of the baselines: SGD and Adam. This is shown using several datasets. Furthermore, the proposed method does not produce a significant overhead with respect to the regular operation of the baselines.
As a main conclusions, the main topic of the paper is relevant to ICLR and the proposed method shows a contribution, although minor, to the operation of two of the most popular optimization methods used today. As a disclaimer, this research area is not close to my main topics of expertise.

---

> ### Author Response · Authors · 2020-11-13
> **thanks for the positive review**
>
> Thanks for the positive review; indeed it's a minor modification, based on sound theory.

---

### Author Response · Authors · 2020-11-13
**summary**

We thank the reviewers for the reviews and constructive comments.

We would like to emphasize that our contribution, to the best of our knowledge, is the first to explore regularities in parameter tensors of neural networks for optimization, and how reformulation of the gradient via new metrics can prefer these patterns.  Our choice of channel-directed Sobolev metrics and variants is one way to favor channel-directed correlation. We hope that future research would explore other patterns in the parameter tensor, and new metrics to derive new optimization methods.

---

### Decision · Program_Chairs · 2021-01-07
**Final Decision**

**Decision:**

Reject

**Comment:**

The paper begins with an observation in standard trained CNNs that the correlations in the output channels are high. Building upon this the paper proposes a new "optimizer" which modifies the gradients to encourage corelations among output channels. They provide a theoretical foundation for the method, by deriving the gradient through placing a riemannian metric on the manifold of parameter tensors which encourages smoothness along the output channel dimension. Two variants (one based on a Sobolev metric) are proposed and are experiments are provided. The underlying idea and the derivation of the gradients were generally appreciated by the reviewers. However some reviewers maintained their concern regarding the effectiveness of the performed experimentation. The gains demonstrated are relatively small over the baselines and more importantly the baselines are quite far off the state of the art baselines for the particular problems. This is the primary reason for my recommendation as experiments are the only source of understanding whether the method is effective (there is little theory - mostly at an intuitive level to justify the form of the optimizer). Overall, I strongly encourage the authors to explore the idea further and strengthen the paper with stronger baselines (perhaps on larger datasets) and resubmit.